# Drug Repositioning Based on the Enhanced Message Passing and Hypergraph Convolutional Networks

**DOI:** 10.3390/biom12111666

**Published:** 2022-11-10

**Authors:** Weihong Huang, Zhong Li, Yanlei Kang, Xinghuo Ye, Wenming Feng

**Affiliations:** 1School of Informatics Science and Technology, Zhejiang Sci-Tech University, Hangzhou 310018, China; 2Zhejiang Province Key Laboratory of Smart Management & Application of Modern Agricultural Resources, School of Information Engineering, Huzhou University, Huzhou 313000, China; 3Department of General Surgery, The First Affiliated Hospital of Huzhou University, Huzhou 313000, China

**Keywords:** drug repositioning, enhanced message passing, hypergraph convolutional network, node and edge embeddings

## Abstract

Drug repositioning, an important method of drug development, is utilized to discover investigational drugs beyond the originally approved indications, expand the application scope of drugs, and reduce the cost of drug development. With the emergence of increasingly drug-disease-related biological networks, the challenge still remains to effectively fuse biological entity data and accurately achieve drug-disease repositioning. This paper proposes a new drug repositioning method named EMPHCN based on enhanced message passing and hypergraph convolutional networks (HGCN). It firstly constructs the homogeneous multi-view information with multiple drug similarity features and then extracts the intra-domain embedding of drugs through the combination of HGCN and channel attention mechanism. Secondly, inter-domain information of known drug-disease associations is extracted by graph convolutional networks combining node and edge embedding (NEEGCN), and a heterogeneous network composed of drugs, proteins and diseases is built as an important auxiliary to enhance the inter-domain message passing of drugs and diseases. Besides, the intra-domain embedding of diseases is also extracted through HGCN. Ultimately, intra-domain and inter-domain embeddings of drugs and diseases are integrated as the final embedding for calculating the drug-disease correlation matrix. Through 10-fold cross-validation on some benchmark datasets, we find that the AUPR of EMPHCN reaches 0.593 (T1) and 0.526 (T2), respectively, and the AUC achieves 0.887 (T1) and 0.961 (T2) respectively, which shows that EMPHCN has an advantage over other state-of-the-art prediction methods. Concerning the new disease association prediction, the AUC of EMPHCN through the five-fold cross-validation reaches 0.806 (T1) and 0.845 (T2), which are 4.3% (T1) and 4.0% (T2) higher than the second best existing methods, respectively. In the case study, EMPHCN also achieves satisfactory results in real drug repositioning for breast carcinoma and Parkinson’s disease.

## 1. Introduction

The process of developing a new drug typically continues for 15–20 years with high costs, less success rate and huge risk from the idea identification to the marketing approval [1]. Drug repositioning, an important part of drug development, refers to the discovery of new and reliable indications for existing drugs. Drug repositioning enables pharmaceutical researchers to save time and reduce costs, thus becoming an effective strategy for the development of new drugs [2,3]. However, drug repositioning still requires intensive study due to unknown and potentially complex pharmacology and biology.

Currently, the methods of drug repositioning mainly include the network-based diffusion analysis method, the machine learning-based method, and the method based on deep learning. Martinez et al. [4] proposed a method called DrugNet concerning network-based diffusion analysis to find new uses for existing drugs. DrugNet implements drug-disease and disease-drug prioritization by propagating information in a heterogeneous network. Wang et al. [5] proposed a TL_HGBI algorithm that integrates similarity and association data about diseases and drugs for predicting drug-related targets. Luo et al. [6] advised an MBiRW method, which applies comprehensive similarity measures and a bi-random walk algorithm to the drug-disease association prediction. In addition, genomics is also combined with network-based methods for drug repositioning. The Connectivity Map (CMap) can be defined as a combination of genome-wide transcriptional expression data that helps in revealing functional connections between drugs, genes, and diseases [7]. Jiang et al. [8] used CMap data to build networks for different types of cancers to identify relationships between small molecules and miRNAs in human cancers and finally came up with therapeutic potentials and new indications for existing drugs.

Although the network-based method has good interpretability, its performance still requires improvement [9]. Machine learning techniques have widely been used in developing more accurate prediction models of drug-disease associations. For example, Gottlieb et al. [10] built the Naive Bayes model to infer the drug indication using the side effects as features. Yang et al. [11] utilized multiple similarity measures of drugs and diseases as features and used the logistic regression model for potential drug indication prediction. However, these feature-based classification methods rely heavily on the feature extraction and selection of negative samples. Recently, matrix factorization and completion methods have been popular in drug repositioning as a result of the flexibility in integrating prior information. For instance, Zhang et al. [12] suggested a matrix factorization method named SCMFDD, which incorporates drug features and disease semantic information into the matrix factorization approach for drug-disease association prediction. Luo et al. [13] recommended a drug repositioning method called DRRS for the new drug indication prediction. It assumes that the drug-disease matrix is low-rank and applies a fast singular value thresholding (SVT) algorithm to complete the drug-disease matrix. Yang et al. [14] used a matrix complementation algorithm called bounded nuclear norm regularization (BNNR) to construct a low-rank drug-disease association matrix approximation consistent with known associations. Although these matrix decomposition and complementation methods have been prevalent, there is still a challenge in the deployment of large-scale network data due to the high complexity of matrix operation.

Deep learning methods have also been successfully applied in drug-disease association prediction. For example, Pham et al. [15] used graph neural networks and multi-head attention mechanisms to predict differential gene expression profiles perturbed by de novo chemicals, which can be applied to drug repurposing in COVID-19. Zeng et al. [16] proposed a network-based deep learning method named deepDR, which uses a multimodal deep autoencoder to fuse positive pointwise mutual information (PPMI) matrices computed from drug-related networks, and then infers drug-disease associations by using a collective variational autoencoder. Recently, graph convolutional networks (GCN) [17] have been introduced for drug- and disease-related association prediction. For instance, Li et al. [18] suggested a NIMCGCN method, which uses GCN to extract the embeddings of miRNAs and diseases from miRNA and disease similarity networks, respectively, and brings a neural inductive matrix completion method to predict miRNA–disease associations. Li et al. [19] constructed an integrated model incorporating GCN, CNN and a channel attention mechanism named GCSENet, which enables the effective implementation of miRNA-disease association prediction. Wang et al. [20] introduced a triple association network including drugs, proteins, and diseases and achieved the fusion of inter-domain information using a bipartite graph convolutional network method (BiFusion), but the network ignored the intra-domain features for the prediction. Yu et al. [21] recommended a method called LAGCN for drug-disease association prediction. It integrates known drug-disease correlations, drug-drug similarity and disease-disease similarity into a heterogeneous network and learns drug and disease embeddings for the heterogeneous network using GCN. Cai et al. [22] applied a bilinear aggregator and GCN to form a drug-disease inter-domain feature extraction module based on the drug and disease similarity networks by considering various network topology information. Ultimately, the inter-domain and intra-domain embeddings are fused in parallel to obtain the drug and disease embedding representation. Although GCN achieves satisfactory performance for the drug-disease association, it has difficulty in capturing the higher-order features of nodes in the graph network. The hypergraph convolution model [23], on the other hand, can effectively solve this problem and has drawn wide attention in recent years.

In order to effectively extract information about the higher-order feature of nodes in the drug- and disease-related network, we propose a new drug repositioning method based on the enhanced message passing and hypergraph convolutional networks (EMPHCN). The contributions of this work are summarized as follows: (1) a multimodal heterogeneous network is built to reveal association information between drugs and diseases by connecting drug-disease association, drug-drug similarity, disease-disease similarity, drug-protein association, protein-protein interaction, and disease-protein association. The foundation for enhanced message passing is provided by this related knowledge. (2) The designed EMPHCN is applied to explicitly model the higher-order associations within the respective domains of drugs and diseases by hypergraph convolution for extracting the intra-domain information. And it uses graph convolutional networks combining node and edge embedding (NEEGCN) to effectively extract the inter-domain information of known drug-disease associations. (3) In addition, it constructs multi-views of multiple drug similarities and extracts the feature of multi-homogeneous information by HGCN combined with the channel attention mechanism, which enhances the intra-domain message passing and forms the intra-domain embedding. Besides, it transfers the feature by a heterogeneous network composed of drugs, proteins, and diseases with GAT and fuses it with the inter-domain feature of known drug-disease associations to form the enhanced inter-domain embedding, and then combines previous intra-domain and inter-domain embeddings to obtain the final embedding of drugs and diseases.

## 2. Materials and Methods

### 2.1. Dataset

The drug-disease associations in dataset 1 (T1) of this paper are selected from the Zhang dataset [12], which contains 18,416 drug-disease associations between 269 drugs and 598 diseases from CTD [24] (http://ctdbase.org/ (accessed on 8 April 2021)). This dataset also collects comprehensive information about drugs such as target, enzyme, drug-drug interaction, pathway and substructure, and these five drug similarity features can be obtained by computation. We select 15,630 drug-disease associations from the 18,416 drug-disease associations in this dataset. Dataset 2 (T2) of this paper is obtained from 1921 known drug-disease relationships for therapeutic indications collected from 6677 approved indications in the repoDB database [25], where we consider only FDA-approved small-molecule drugs. We obtained 10 drug similarity features by referring to the method proposed by Zeng et al. [16]. In order to fuse more drug features for enhancing the intra-domain message passing of drug, we input various drug similarity information (e.g., target, enzyme, drug-drug interaction, and pathway) in T1 as well as T2 (e.g., clinical similarity, drug side effects’ similarity, and chemical similarity, where the chemical similarity is computed by Tanimoto score [26], which can be obtained by the Chemical Development Kit [27] according to the SMILES string [28] of drugs) into the hypergraph convolutional network as multiple similarity matrices of drugs, respectively.

Although datasets T1 and T2 contain a limited number of disease and drug nodes, they contain a massive amount of protein association information which is helpful for the predictions of drug-disease associations. In addition, in order to test the robustness of our baseline model, we introduce 2 public datasets, Fdataset [10] and Cdataset [6], which do not contain the protein association network and various drug characteristics. Fdataset contains 593 drugs, 313 diseases and 1933 confirmed drug-disease associations. Cdataset includes 2532 associations between 663 drugs and 409 diseases.

The protein association networks (drug-protein, disease-protein, protein-protein interaction (PPI)) in this paper draw data from the databases used by Wang et al. [20] The DGIdb database [29] (https://dgidb.genome.wustl.edu (accessed on 9 September2021)) is a drug-gene interaction database, which integrates not only drug-gene interactions reported in the existing literature, but also records drug-gene interactions in more than 30 databases such as DrugBank [30], PharmGKB [31], Chembl [32], and TTD [33], providing information on genes and their known or potential drug associations. We pull target protein-coding genes of a given drug from DGIdb, and then map genes to proteins according to gene names and obtain the drug-protein association. At least 2 targets are required for each drug which is motivated by the notion that a drug is utilized to treat a distinct disease, most likely due to its off-target activities.

The disease-protein associations are extracted from DisGeNET [34] (https://www.disgenet.org (accessed on 5 May 2021)), where we extract the protein-coding genes of specific diseases and then map them to the corresponding protein. It is required that at least 1 protein is associated with each disease which makes it more biologically meaningful. The drug name is represented by the DrugBank ID, and the common name of each disease is annotated according to MeSH [35] (http://www.ncbi.nlm.nih.gov/ (accessed on 8 April 2021)), which can be converted to UMLS ID [36] using DisGeNET to discover the related protein.

For the PPI network, we refer to the human PPI information compiled by Menche et al. [37] to extract the protein-protein associations; that is, the initial weight of each edge is set to 1, and one protein is associated with at least 1 protein. Ultimately, we obtain the multi-association networks by combining the above data, as shown in Table 1.

### 2.2. Multi-Association Network Construction

We apply the multiple similarity matrices of the drugs mentioned in the previous section to construct a drug multi-view network; namely, the set of multiple similarity matrices of drugs is denoted as Pr=P1r,P2r,…,PSr, PSr∈ℝM×M, where *M* is the number of drugs, and *S* represents the total number of the similarity matrices. And then, it is used to construct the multi-view network as Gr={G1r,G2r,…,GSr}, in which the set of its adjacency matrices is denoted as Ar=Pr=P1r,P2r,…,PSr.

The similarity matrix of diseases is expressed as Sd∈ℝN×N, where *N* denotes the number of diseases. The similarity between diseases is obtained from the MeSH database [35], in which diseases are classified into various categories. We construct a directed acyclic graph (DAG) to compute the semantic similarities of diseases based on the reference of Wang et al. [38]. The disease-disease similarity network is denoted as Gd and its adjacency matrix is Ad=Sd∈ℝN×N.

Here, the known drug-disease association network is represented as a graph $G_{rd}$ with the adjacency matrix $A^{rd}\in {\left\{0,1\right\}}^{M\times N}$ (If drug $r_{i}$ is associated with disease $d_{j}$, $A_{ij}^{rd}$ = 1. Otherwise, $A_{ij}^{rd}$ = 0). In addition to the above-constructed graphs $G_{r}$, $G_{d}$ and $G_{rd}$, we build the graph structures by using the association data of protein-drug, protein-disease and PPI described in the previous section, which are denoted as $G_{dp},G_{rp},\;\mathrm{and}\;G_{p}$, respectively. The overall association network is shown in Figure 1.

### 2.3. Model Framework

In this section, we introduce a deep learning model named EMPHCN to predict the association information between drugs and diseases. The overall workflow is shown in Figure 2, which includes the intra-domain and inter-domain message passing of drugs and diseases. A multi-view network is constructed according to multiple similarity features of drugs for the intra-domain message passing of drugs, which is enhanced by the combination of HGCN and channel attention mechanism (Figure 2a). Besides, we directly apply HGCN to extract the intra-domain embeddings of diseases. The inter-domain message passing includes 2 parts (Figure 2b): 1 is the inter-domain message passing for known drug-disease associations, which is enhanced by graph convolutional networks with node and edge embedding (NEEGCN), and the other is a heterogeneous network composed of proteins, drugs and diseases, which enriches the message passing according to GAT. The embedding of 2 parts is then summed to obtain the inter-domain feature of drugs and diseases, and ultimately, all intra-domain and inter-domain features are integrated for the prediction of the drug-disease association.

### 2.4. Drug and Disease Intra-Domain Message Passing

We first employ hypergraph convolutional networks (HGCN) [23] in the intra-domain message passing to extract the intra-domain information of drugs and diseases in $G_{r}$ and $G_{d}$, respectively. The general graph network structure is usually represented by an adjacency matrix, where each edge connects only two vertices. Hypergraphs, on the other hand, can be used to encode the higher-order data correlation (beyond pairwise connections) by using their degree-free hyperedges, i.e., hypergraphs with flexible hyperedge properties are easily scalable for multi-modal data.

A hypergraph is usually defined as *G* = (*V*, *E*), where *V* represents the vertex set, and *E* denotes the hyperedge set. An incidence matrix H∈ℝNv×Ne is used to represent connections among vertices on the hyper-graph, where Nv is the number of vertices and Ne is the number of hyperedges. Each element in *H* is defined as
(1)hv,e= 1,   if v∈e 0,   if v∉e

For any vertex v∈V, hv,e=1 when the hyperedge e∈E is associated with vertex v, otherwise hv,e=0. Simultaneously, the degree dv of vertex v and the degree δe of hyperedge e are represented by the diagonal matrix Dv∈ℝNv×Nv and De∈ℝNe×Ne, respectively, then the degree of the vertex and the degree of the hyperedge can be defined as dv=∑e∈Ehv,e, δe=∑v∈Vhv,e.

The incidence matrix *H* and vertex feature *X* can be fed into the hypergraph neural networks with a hypergraph convolutional layer defined as
(2)Xl+1=Dv−1/2HDe−1HTDv−1/2Xlθl
where two diagonal matrices Dv−1/2 and De−1 are used for normalization, θl is the learnable weight and Xl+1 is the feature embedding learned by the HGCN. Each hyperedge in EMPHCN is built by connecting a vertex with its neighboring *K* vertices (Here, we set *K* = 15) based on the adjacency relationship on the graph, so *N* vertices construct *N* hyperedges with the incidence matrix H∈ℝN×N.

Specifically, we use the initial input graph Gr,Gd to construct the hypergraphs respectively, and the corresponding hypergraph incidence matrices are set as Hr and Hd, respectively. Here, Hr=H1r,H2r,…,HSr, HSr∈ℝM×M, Hd∈ℝN×N, *M* is the number of drugs, *S* is the total number of incidence matrices of drugs, and *N* is the number of diseases. We use one of Hr (Here, we chose H1r) and Hd as the initial input Xintra_r0, Xintra_d0 of the model
(3)Xintra_r0Xintra_d0=H1rHd 

According to Equation (2), the updated embeddings in the disease and drug domains can be obtained as $X_{intra\_r}^{l+1},\;X_{intra\_d}^{l+1}$ respectively
(4)$$\;X_{intra\_r}^{l+1}=D_{vr}^{-1/2}H_{r}D_{er}^{-1}H_{r}{}^{T}D_{vr}^{-1/2}\;X_{intra\_r}^{l}\;\theta_{intra\_r}^{l}\;X_{intra\_d}^{l+1}=D_{vd}^{-1/2}H_{d}D_{ed}^{-1}H_{d}{}^{T}D_{vd}^{-1/2}\;X_{intra\_d}^{l}\;\theta_{intra\_d}^{l}$$

From Equation (4), we can calculate the multi-channel drug embedding Xintra_rl+1=[Xintra_r1l+1,Xintra_r2l+1,…,Xintra_rSl+1], Xintra_rl+1∈ℝM×F×S by the hypergraph convolution of each incidence matrix of drugs and similarly obtain the disease embedding Xintra_dl+1∈ℝN×F. Here, F is the dimensionality of the embedding.

Furthermore, in order to utilize the multivariate feature information of drugs for the intra-domain message passing enhancement, we employ the attention mechanism ECA-Net [39] to learn the importance of each channel in $X_{intra\_r}^{l+1}$.

The specific operation process is as follows: we first use the global average pooling on the spatial dimension to perform the feature compression and get the new embedding Zr∈ℝ1×1×S,
(5)Zr=gXintra_rl+1
where gx=1F×M∑i=1,j=1F,Mxij is the global average pooling (GAP), and then we further compute the attention factor ωr corresponding to each channel, and in order to reduce the model complexity, we use a 1D convolution operation to realize that all channels share the same parameters
(6)ωr=σconv1dkZr
where ωr=[ω1r,ω2r,…,ωSr],ωr∈ℝ1×S, conv1d stands for the 1D convolution, *k* represents the convolution kernel size (Here, we set it as *k* = 3 × 1), σ it the Sigmoid activation function.

Ultimately, we sum up the features of all channels to get X˜intra_rl+1∈ℝM×F, which is used as the drug embedding of the next layer of network input
(7)X˜intra_rl+1=∑i=1SωirXintra_ril+1

### 2.5. Drug and Disease Inter-Domain Message Passing

Drug-disease inter-domain message passing contains the known drug-disease association network, which is treated as the main part of message passing. Moreover, we enhance the inter-domain message passing by introducing a heterogeneous network composed of proteins, drugs and diseases.

### 2.6. Known Drug-Disease Inter-Domain Message Passing

GCN has widely been used in the recommendation system and the prediction model [40]. However, it usually only considers the node embedding in the graph convolutional process while ignoring the role of edges [41]. We introduce a novel graph convolution model (NEEGCN) for bipartite graph link prediction, which can combine the embedding of edges and nodes. We first map the edge information into the node domain to obtain the corresponding drug node feature $X_{r\_e}^{l}$ and disease node feature $X_{d\_e}^{l}$, respectively,
(8)$$X_{r\_e_{i}}^{l}=\overset{N}{\underset{j}{\sum }}A_{ij}^{rd}w_{r\_e}^{l}\;,\;X_{d\_e_{j}}^{l}=\overset{M}{\underset{i}{\sum }}{\left(A_{ij}^{rd}\right)}^{T}w_{d\_e}^{l}\;$$
where $A^{rd}$ denotes the adjacency matrix of the known drug-disease association graph network, $W_{r\_e}^{l}$ and $W_{d\_e}^{l}$ are the trainable weight matrices. We refer to the intra-domain initial input in Equation (3) above as the initial inter-domain input, the edge information mapped to the drug and disease node embeddings is then fused with the original drug and disease node embeddings, and the message passing between the nodes of drugs and diseases is defined as
(9)$$X_{inter\_r_{i}}^{l+1}=\underset{\left(d_{j}\right)\in N\left(r_{i}\right)}{\sum }\frac{1}{\sqrt{\left|N\left(d_{j}\right)\right|\left|N\left(r_{i}\right)\right|}}(X_{inter\_d_{j}}^{l}⨀X_{d\_e_{j}}^{l})W_{d\to r}^{\left(l\right)}\;\;X_{inter\_d_{j}}^{l+1}=\underset{\left(r_{i}\right)\in N\left(d_{j}\right)}{\sum }\frac{1}{\sqrt{\left|N\left(d_{j}\right)\right|\left|N\left(r_{i}\right)\right|}}(X_{inter\_r_{i}}^{l}⨀X_{r\_e_{i}}^{l})W_{r\to d}^{\left(l\right)}\;$$
where $N\left(r_{i}\right)$ and $N\left(d_{j}\right)$ are the set of inter-domain neighbors of node $r_{i}$ and node $d_{j}$ respectively, and $\frac{1}{\sqrt{\left|N\left(d_{j}\right)\right|\left|N\left(r_{i}\right)\right|}}$ is the regularization form of GCN, which can avoid the random embedding explosion with propagation. We combine edge features with node features by the element-wise multiplication of 2 vectors. $W_{d\to r}^{\left(l\right)}$ and $W_{r\to d}^{\left(l\right)}$ are trainable weights for projecting the node embedding from 1 domain to another.

### 2.7. Protein-Related Inter-Domain Message Passing

In addition to using the known drug-disease network for the message passing, we add a heterogeneous network composed of proteins, diseases and drugs and combine them to form an inter-domain message-passing enhancement module for extracting the inter-domain features of drugs and diseases. We refer to the method suggested by Wang et al. [20] and achieve message-passing between them by the graph attention network (GAT) [42], which can help us score a vast amount of protein-association information to filter out the important association information. The propagation process is first implemented as
(10)$$\left\{\begin{array}{c} E_{r\to p}^{l}={\mathrm{GAT}}_{r\to p}\left(X_{r}^{l}\right)\\ E_{d\to p}^{l}={\mathrm{GAT}}_{d\to p}\left(X_{d}^{l}\right)\\ E_{p}^{l}=concat\left(E_{r\to p}^{l},E_{d\to p}^{l}\right) \end{array}\right.\;$$
where $X_{r}^{l}\;\mathrm{and}$ $X_{d}^{l\;}$ are the disease and drug embeddings of the lth layer and $E_{r\to p}^{l}$, $E_{d\to p}^{l}$ are the information from drug to protein through GAT and the information from disease to protein through GAT in the lth layer, respectively, and we fuse them to get the protein embedding $E_{p}^{l}$. Here, GAT is used to achieve the projection of node information from the v-domain to the u-domain, and the updated formula is
(11)$${\mathrm{GAT}}_{v\to u}:{\vec{\;x}}_{u_{i}}=ReLU(\sum_{v_{j}\in N_{u_{i}}}\;\alpha_{u_{i},\nu_{j}}W^{v}{\vec{x}}_{v_{j}})$$
where $W^{v}$ is a trainable weight matrix, ${\vec{x}}_{v_{j}}$ represents the feature of the node and $\nu_{j}$, $\alpha_{u_{i},\nu_{j}}$ is the attention weight coefficient as
(12)$$\alpha_{u_{i},\nu_{j}}=\frac{\exp(\rho ({\vec{a}}^{T}\left[W^{u}{\vec{x}}_{u_{i}}∥W^{v}{\vec{x}}_{v_{j}}\right]))}{\sum_{v_{j}\in N_{u_{i}}}exp(\rho ({\vec{a}}^{T}\left[W^{u}{\vec{x}}_{u_{i}}∥W^{v}{\vec{x}}_{v_{j}}\right]))}\;$$
where ρ is the LeakyReLU activation function, $\vec{a}$ denotes the weight vector, and ∥ is the concatenation operation.

Then, in order to enable smooth features between protein neighborhood nodes, a single layer of GAT is applied to perform the protein intra-domain message passing by the PPI network
(13)$${\tilde{E}}_{p}^{l}={\mathrm{GAT}}_{p\to p}(E_{p}^{l})$$

We project the updated protein node embedding ${\tilde{E}}_{p}^{l}$ back to the drug and disease domains to obtain the drug embedding $X_{p\_r}^{l}$ and the disease embedding $X_{p\_d}^{l}$, respectively.
(14)$$\left\{\begin{array}{c} X_{p\_r}^{l}={\mathrm{GAT}}_{p\to r}\left(E_{p}^{l}\right)\\ X_{p\_d}^{l}={\mathrm{GAT}}_{p\to d}\left({\tilde{E}}_{p}^{l}\right) \end{array}\right.$$

### 2.8. Association Prediction for Drugs and Diseases

After computing the respective intra-domain and inter-domain features of drugs and diseases, we fuse these embeddings of drugs and diseases as
(15)$$\;X_{r}^{l+1}=X_{intra\_r}^{l}+X_{inter\_r}^{l}+X_{p\_r}^{l}\;X_{d}^{l+1}=X_{intra\_d}^{l}+X_{inter\_d}^{l}+X_{p\_d}^{l}$$
where $X_{r}^{l+1}$, $X_{d}^{l+1}$ are the embeddings of the next layer of drugs and diseases, respectively. We can similarly obtain $X_{r}^{l+2}$ and $X_{d}^{l+2}$, and then combine them via a skip connection to get the final embeddings of drugs and diseases ${\hat{X}}_{r},{\hat{X}}_{d}$ respectively
(16)$${\hat{X}}_{r}=X_{r}^{l+1}+X_{r}^{l+2},\;{\hat{X}}_{d}=X_{d}^{l+1}+X_{d}^{l+2}$$

Lastly, we obtain the final association information between drugs and diseases by the matrix multiplication with the following equation
(17)$${\hat{A}}^{rd}=sigmoid({\hat{X}}_{r}{\hat{X}}_{d}{}^{T})$$
where the matrix ${\hat{A}}^{rd}$ is the predicted score probability matrix, and the values in ${\hat{A}}^{rd}$ represent the probability of drug-disease association.

### 2.9. Optimization and Parameter Setting

Since the number of experimentally confirmed associations is much smaller than the number of drug-disease pairs, and because the sparsity of various drug-disease datasets is inconsistent, we adopt the weighted binary cross-entropy as a loss function to balance the positive and negative sample ratios. We denote known drug-disease association pairs as positive samples and other pairs as negative samples, and the loss function is defined as follows
(18)$$\;\mathrm{Loss}=-\frac{1}{N\times M}(\lambda \times \sum_{\left(i,j\right)\in y}logA_{ij}^{rd}+\sum_{\left(i,j\right)\in \bar{y}}\left(1-logA_{ij}^{rd}\right))$$
where $\bar{y}$ and y are the number of negative and positive samples, respectively, and the balance factor is λ = $\left|\bar{y}\right|$/|y|. We use the Adam optimizer [43] to minimize the loss function.

The EMPHCN is divided into two hidden layers, and the main parameters of this framework include the first and second hidden layer embedding dimensions, $k_{1}$ = 256, $k_{2}$ = 128, the initial learning rate of the optimizer, *lr* = 0.002, the regular dropout rate, *β* = 0.4. According to the sparsity of the dataset, we set the edge dropout rate *γ* and the training epoch *μ* as $\gamma_{\left(T1\right)}$ = 0.35,$\;\gamma_{\left(T2\right)}$ = 0.1,$\;\mu_{\left(T1\right)}$ = 4500,$\;\mu_{\left(T2\right)}$ = 3200, respectively.

## 3. Results

### 3.1. Ablation Experiment of EMPHCN Model

We conduct the ablation experiment to verify the effectiveness of each part of our EMPHCN model. Specifically, we analyzed the impact of each part in our model according to the results of a 10-fold cross-validation on the dataset T1. The performance of different variants of EMPHCN is shown in Figure 3. We first used the GCN to replace the HGCN and found that the AUC and AUPR values achieved by the GCN were 0.4% and 0.8% lower than those of the HGCN, respectively. For the baseline model EMP_base (with the HGCN and the NEEGCN), we set a single drug similarity matrix in the intra-domain of drugs and only used known drug-disease associations in the inter-domain. Then we introduced the drug multi-view information and integrated the HGCN and ECA-Net to form an intra-domain fusion enhancement module. We found that the AUC and AUPR values using the intra-domain fusion enhancement module were better than those of the baseline method using a single similarity matrix, which both had a 0.3% improvement, as shown in Figure 3. In addition, we introduced additional networks $G_{dp},G_{rp},\;\mathrm{and}\;G_{p}$ into the inter-domain information and combined known drug-disease associations to form the inter-domain fusion enhancement module. We found that the AUC and AUPR values achieved by the inter-domain fusion enhancement module were 0.5% and 0.6% higher than those of the baseline method, respectively, whose inter-domain module is only composed of known drug-disease associations. According to these comparisons, we can clearly see that the EMPHCN using HGCN and intra-domain and inter-domain fusion modules (with the intra-domain and inter-domain message passing enhancement) produced an improvement in the performance of drug repositioning.

### 3.2. Comparison between EMPHCN and Other Methods

In this section, we compare EMPHCN with six state-of-the-art drug repositioning methods listed below. Among these methods, the LAGCN method [21] uses an attention mechanism to combine the embeddings from multiple graph convolutional layers to predict drug-disease associations. The SCMFDD [12] provides a similarity constraint matrix decomposition method for predicting drug-disease correlations. The NIMCGCN [18] applies GCNs to learn the intra-domain information of miRNA and disease from similarity networks and builds a neural inductive matrix completion model to predict miRNA-disease associations. BNNR [14] proposes a bounded nuclear norm regularization method to complete a drug-disease heterogeneous network for drug repositioning. DRRS [13] constructs a drug repositioning recommendation system for novel drug indications by integrating related data sources and validated information on drugs and diseases. The DRHGCN [22] designs inter- and intra-domain feature extraction modules which learn drug and disease embeddings by GCNs, and then parallelly fuse these embeddings for drug and disease association prediction.

We applied 10-fold cross-validation when comparing the EMPHCN with other methods. The results are shown in Table 2 as well as Figure 4a–d. On dataset T1, EMPHCN obtains the highest average AUPR and AUC at 0.593 and 0.887, respectively. On dataset T2, EMPHCN outperforms the other six models with a final average AUC of 0.961, which is 2.3% higher than the second-best method, DRHGCN, and achieves a final average AUPR of 0.526, which is 3.4% higher than the second-best method, BNNR. Because we introduce more homogeneous and heterogeneous networks and apply HGCN to extract the feature in intra-domain as the message passing, EMPHCN can achieve a satisfactory prediction result for the drug repositioning. At the same time, we find that the EMPHCN baseline model (EMP_base) also keeps excellent results in the comparative experiment (Figure 5, Table 2), which achieved the highest average AUC and AUPR values (0.941 and 0.593, respectively) in the benchmark datasets.

We also evaluated the effect of the ratio of positive and negative samples in the training set on the performance of the EMPHCN. We chose the T2 dataset with sparse positive samples and selected positive and negative sample ratios of 1:10, 1:50, 1:80, 1:100, 1:120, and 1:140 (where 1:140 is close to the maximum ratio in the dataset). Besides, we various-ly set a small portion of the positive samples missing as a control group, i.e., set the number of positive samples as {100%, 95%, 90%, and 85%} of the original positive samples, respectively. The test results are shown in Figure 6. We find that the performance of the model is improved as the number of positive samples is increased, probably because more positive samples are beneficial for the message passing in the prediction model. In addition, we find when the ratio of positive samples is in the range of 90% to 100%, the performance of the model is stable and improves slightly with the increase in the proportion of negative samples, probably because of the double influence of the increase of the number of training samples and negative samples.

In addition, we implement the statistically significant difference analysis on the T1 dataset for the AUROC and AUPR results of 10-fold cross-validation between EMPHCN and other methods. Figure 7 demonstrates that when the *p*-value = 0.05, EMPHCN considerably differs from other approaches.

### 3.3. Investigation of Novel Predictions

To validate the ability of diverse models to recover new disease associations corresponding to drugs (i.e., no drug association information for new diseases), we first choose 20% of the disease nodes as the test set, and for each disease in the test set, we delete all known drug-disease associations associated with that disease. Then we use the remaining 80% of the disease nodes corresponding to known drug-disease associations as the training set. This process ensures that the disease nodes in the test sample are unknown new nodes. We repeat the test five times for each prediction model to obtain an average result. The corresponding results, including ROC and PR curves, are shown in Table 3 and Figure 8a–d. We find that EMPHCN achieved excellent performance on datasets T1 and T2 (AUC = 0.806 on T1, 0.845 on T2), which were superior to other state-of-the-art prediction methods.

In addition, we further validate the number of positive samples successfully recovered from the first k candidate samples. Considering the various number of positive samples in various datasets, we set the threshold k to 3000 (T1) and 300 (T2), respectively. Under the T1 dataset, we find that the number of positive samples successfully recovered by EMPHCN and SCMFDD is significantly higher than those of other models, as shown in Figure 9a. For the T2 dataset, the number of positive samples successfully recovered by EMPHCN is significantly higher than in other models, as shown in Figure 9b, which indicates again that EMPHCN has the outstanding ability to prioritize potential disease-related drugs.

### 3.4. Case Study

To further verify the reliability and capability of the drug-disease association prediction model, we applied the EMPHCN to predict candidate drugs for two diseases, including breast carcinoma and Parkinson’s disease (PD). When identifying potential drugs for breast carcinoma and Parkinson’s disease, we used all known drug-disease associations in T2 as a training set to predict whether a missing drug-disease association existed and ranked the predicted drug candidates according to the magnitude of their probability values.

The top 10 drug candidates for breast carcinoma predicted by the EMPHCN are listed in Table 4. We found that 10 of these (100% success rate) are confirmed by various published pieces of evidence. For instance, topotecan is an antineoplastic agent used to treat ovarian cancer, small-cell lung cancer or cervical cancer. A previous study reports that primary chemotherapy with topotecan is an effective and well-tolerated treatment for patients with breast cancer and central nervous system (CNS) metastases [44]. Herein, topotecan is the first predicted candidate for potentially treating breast carcinoma. Carboplatin, an organoplatinum antineoplastic alkylating agent, used in the treatment of advanced ovarian carcinoma, is predicted by EMPHCN to be associated with breast carcinoma. The study by Martin et al. [45] indicates that carboplatin is an active drug in metastatic breast cancer (MBC) patients without previous exposure to chemotherapy. Campana et al. [46] observed that elderly breast cancer patients were highly responsive to the toxicity of electrochemotherapy (ECT) and achieved durable local tumor control, where the ECT treatment schedule consisted of intravenous or intratumoral bleomycin followed by locally-delivered electric pulses.

For Parkinson’s disease, the top 10 candidate drugs predicted by EMPHCN are listed in Table 5. We find 8 of which (80% success rate) were confirmed by various literature evidence. For example, pergolide is an ergot-derived dopamine receptor agonist utilized to treat Parkinson’s disease and restless legs syndrome. The study by Mizuno et al. [54] confirms that pergolide has efficacy in patients with Parkinson’s disease, either as monotherapy or in combination with levodopa. Galantamine is a cholinesterase inhibitor used to manage mild to moderate dementia associated with Alzheimer’s disease. Aarsland et al. [55] report that galantamine is approved to be serviceable for patients with Parkinson’s disease.

In addition, we used the same model EMPHCN to predict the top five diseases likely to be associated with felodipine, as shown in Table 6. Lorimer et al. [61] verified that felodipine has, overall, a modest but significant anti-anginal benefit when combined with a beta-blocker.

## 4. Discussion

In this paper, we propose the EMPHCN, a novel prediction model to recover potential drug-disease associations based on enhanced message passing and hypergraph convolutional networks. Compared with existing methods, the EMPHCN incorporates multiple drug similarities and enhances the intra-domain message passing by combining hypergraph convolution with channel attention mechanism, and improves the inter-domain message passing by incorporating protein, drug and disease association networks, which can outperform other drug-disease association prediction methods.

Although the EMPHCN can be chosen as a powerful tool for predicting drug-disease association information, there are some issues that can be improved in our future work. For instance, we will consider more biological entities, such as miRNAs and microorganisms involved in drug-disease associations, to construct a heterogeneous network with more types of entities and links for a better understanding of drug and disease features for drug repositioning. Because our current model does not involve genomics, transcriptomics, other omics and cell line data from an individual patient, we will extend our method by combining with the multi-omics information to prescribe the patient a specifically repurposed drug in our future work. In addition, we currently apply the PPI information to construct the association network for training. In our future work, we will use the protein similarities to construct the protein-protein association network to research the relationship between the off-target consequences and the repositioning opportunities.

## Figures and Tables

**Figure 1 biomolecules-12-01666-f001:**
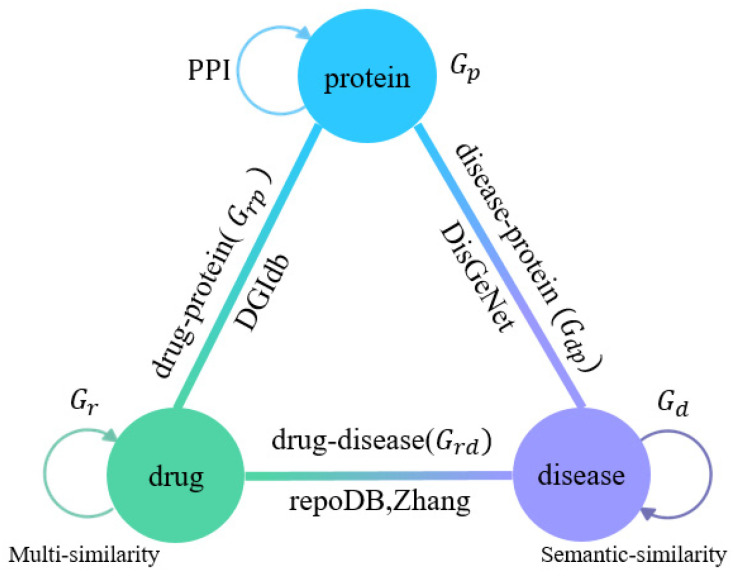
Overall network construction for drug-disease association prediction.

**Figure 2 biomolecules-12-01666-f002:**
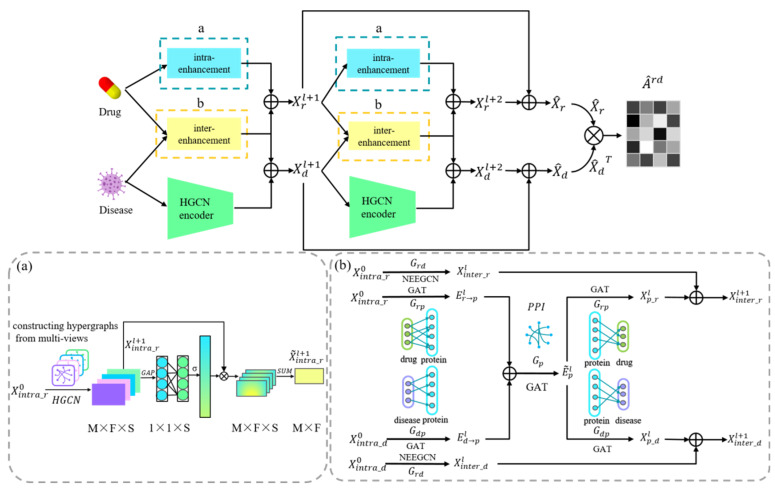
EMPHCN framework diagram. (**a**) Intra-domain message passing enhancement of drugs. (**b**) Inter-domain message passing enhancement of drugs and diseases.

**Figure 3 biomolecules-12-01666-f003:**
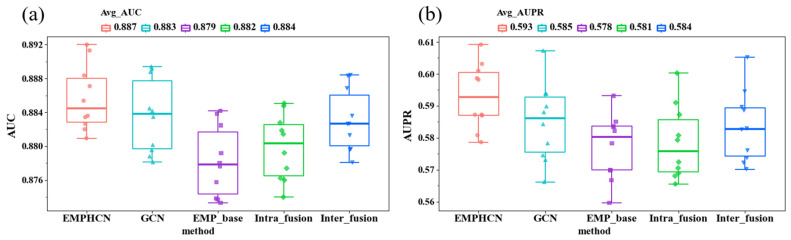
Comparison with 10-fold cross-validation for different variants of EMPHCN on dataset T1. (**a**) AUC comparison of different variants of EMPHCN. (**b**) AUPR comparison of different variants of EMPHCN.

**Figure 4 biomolecules-12-01666-f004:**
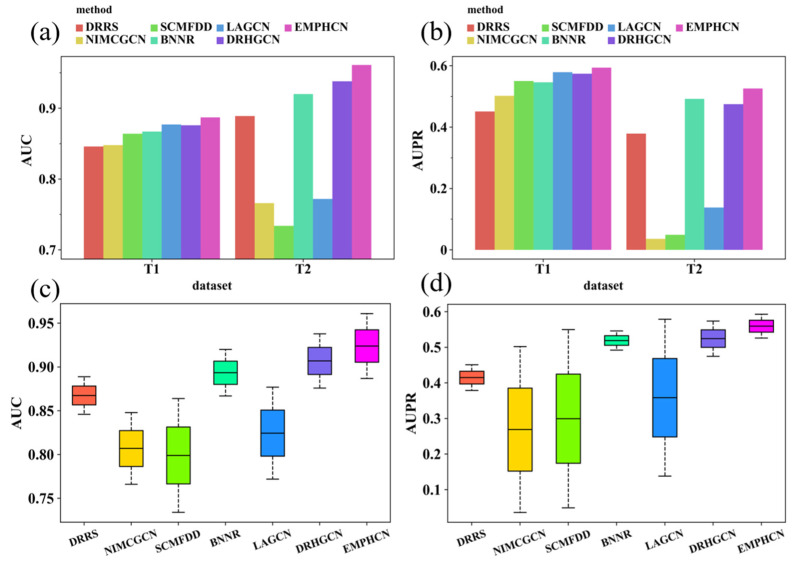
Performance comparison with all methods by 10-fold-cross-validation on datasets T1 and T2. (**a**) AUC comparison of different methods. (**b**) AUPR comparison of different methods. (**c**) Box-plot comparison of average AUCs (**d**) Box-plot comparison of average AUPRs.

**Figure 5 biomolecules-12-01666-f005:**
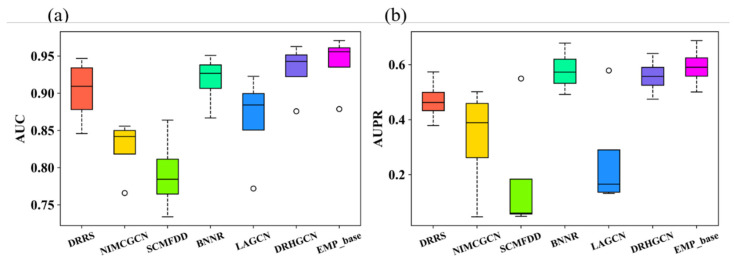
Performance of the baseline model comparison with all methods by 10-fold-cross-validation on four benchmark datasets. (**a**) Average AUC comparison of different methods. (**b**) Average AUPR comparison of different methods.

**Figure 6 biomolecules-12-01666-f006:**
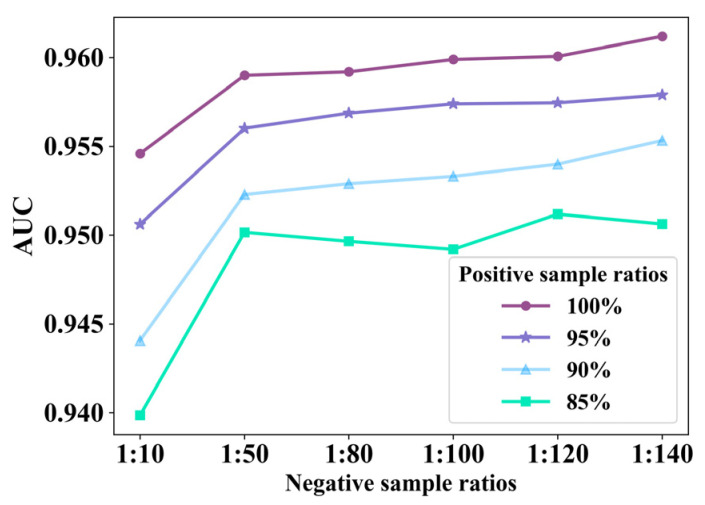
Results of EMPHCN under different negative sample ratios and different positive sample ratios on dataset T2.

**Figure 7 biomolecules-12-01666-f007:**
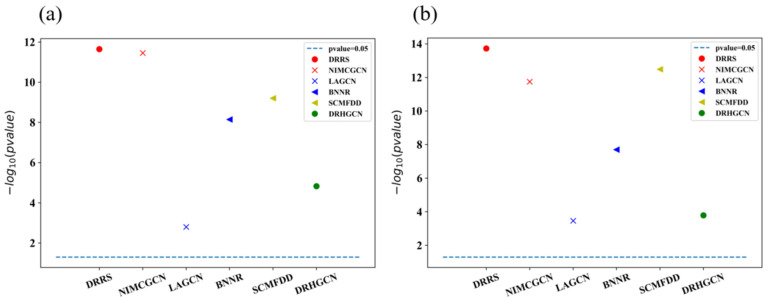
Statistically significant difference analysis of 10-fold cross-validation between EMPHCN and other methods on the T1 dataset. Since the *p*-value calculated by various models are quite unlike, we provide the result of ${-\log}_{10}$ (*p*-value) instead of the *p*-value. (**a**) Significance analysis of AUC results. (**b**) Significance analysis of AUPR results.

**Figure 8 biomolecules-12-01666-f008:**
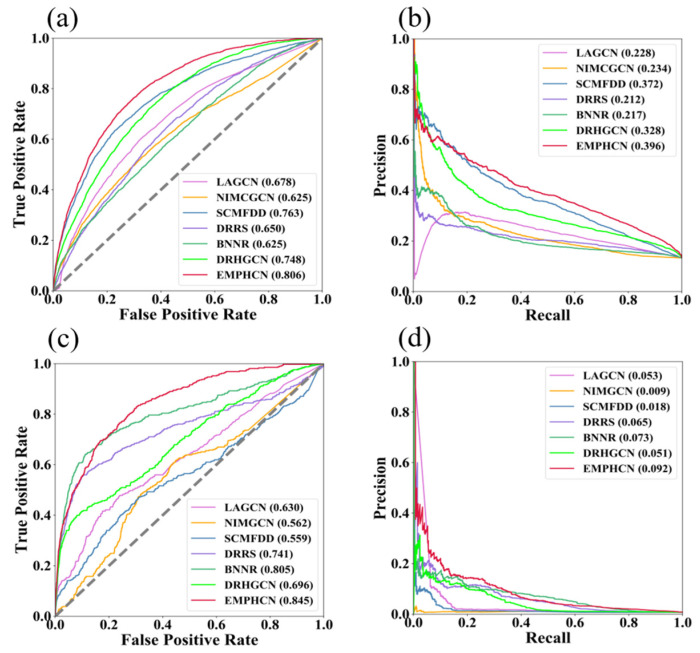
Comparison of novel prediction results with different methods. (**a**) ROC curve comparison of various methods on the T1 dataset. (**b**) PR curve comparison of various methods on the T1 dataset. (**c**) ROC curve comparison of various methods on the T2 dataset. (**d**) PR curve comparison of various methods on the T2 dataset.

**Figure 9 biomolecules-12-01666-f009:**
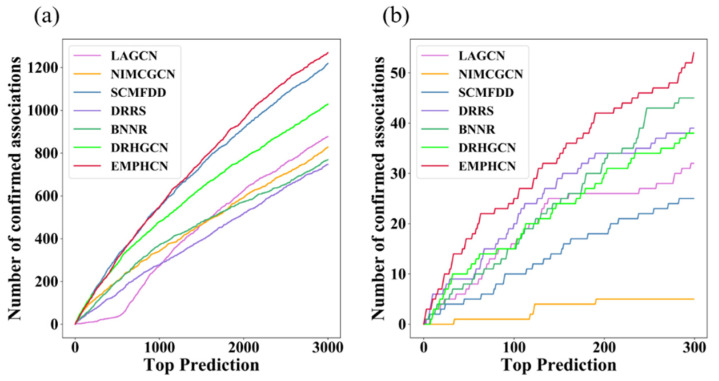
Comparison of novel prediction results with different methods. (**a**) Comparison of the number of successfully recovered positive samples on dataset T1. (**b**) Comparison of the number of successfully recovered positive samples on dataset T2.

**Table 1 biomolecules-12-01666-t001:** Data information on our network construction for drug repositioning.

Dataset	P	Drug	Disease	Protein	Domain	Interaction	Sparsity
T1	4	263	480	6059	drug-disease	15,630	0.1238
					drug-protein	5620	
					disease-protein	20,019	
					protein-protein	49,406	
T2	3	850	339	5105	drug-disease	1921	0.0067
					drug-protein	10,872	
					disease-protein	12,625	
					protein-protein	41,903	
Cdataset	1	663	409	----------	drug-disease	2532	0.0093
Fdataset	1	593	313	----------	drug-disease	1933	0.0104

Note: P denotes the number of similarity matrices for drugs in various datasets. Sparsity represents the ratio of the number of known associations to the number of all possible associations.

**Table 2 biomolecules-12-01666-t002:** Result comparison of 10-fold cross-validation experiments with different methods on benchmark datasets.

Dataset	DRRS	NIMCGCN	SCMFDD	BNNR	LAGCN	DRHGCN	EMP_base	EMPHCN
AUROC								
T1	0.846 ± 0.002	0.848 ± 0.004	0.864 ± 0.002	0.867 ± 0.001	0.877 ± 0.002	0.876 ± 0.001	0.879 ± 0.001	0.887 ± 0.002
T2	0.889 ± 0.002	0.766 ± 0.002	0.734 ± 0.004	0.920 ± 0.001	0.772 ± 0.003	0.938 ± 0.002	0.958 ± 0.003	0.961 ± 0.003
Cdataset	0.947 ± 0.002	0.856 ± 0.004	0.794 ± 0.001	0.951 ± 0.001	0.923 ± 0.004	0.963 ± 0.001	0.971 ± 0.001	---------
Fdataset	0.930 ± 0.002	0.836 ± 0.004	0.775 ± 0.001	0.934 ± 0.001	0.892 ± 0.003	0.948 ± 0.002	0.954 ± 0.002	---------
Avg	0.903	0.827	0.792	0.918	0.866	0.931	0.941	---------
AUPR								
T1	0.451 ± 0.002	0.502 ± 0.004	0.550 ± 0.002	0.546 ± 0.001	0.579 ± 0.002	0.574 ± 0.001	0.578 ± 0.001	0.593 ± 0.003
T2	0.379 ± 0.002	0.047 ± 0.002	0.049 ± 0.003	0.492 ± 0.001	0.138 ± 0.004	0.475 ± 0.001	0.501 ± 0.003	0.526 ± 0.002
Cdataset	0.574 ± 0.003	0.445 ± 0.002	0.060 ± 0.002	0.679 ± 0.001	0.194 ± 0.002	0.655 ± 0.002	0.688 ± 0.002	---------
Fdataset	0.475 ± 0.006	0.354 ± 0.005	0.062 ± 0.002	0.601 ± 0.001	0.134 ± 0.002	0.566 ± 0.002	0.604 ± 0.002	---------
Avg	0.470	0.337	0.180	0.580	0.261	0.568	0.593	---------

**Table 3 biomolecules-12-01666-t003:** The results of novel predictions on datasets T1 and T2 by different methods.

Dataset	Methods	AUPR	AUC	RE	ACC	F1
T1	LAGCN	0.228	0.678	0.503	0.756	0.325
	NIMCGCN	0.234	0.625	0.438	0.757	0.289
	SCMFDD	0.372	0.763	0.531	0.845	0.417
	DRRS	0.212	0.650	0.555	0.662	0.295
	BNNR	0.217	0.625	0.580	0.582	0.269
	DRHGCN	0.328	0.748	0.539	0.788	0.368
	EMPHCN	0.396	0.806	0.547	0.859	0.442
T2	LAGCN	0.053	0.630	0.098	0.986	0.094
	NIMCGCN	0.009	0.562	0.474	0.689	0.022
	SCMFDD	0.018	0.559	0.067	0.987	0.072
	DRRS	0.065	0.741	0.249	0.981	0.156
	BNNR	0.073	0.805	0.165	0.987	0.153
	DRHGCN	0.051	0.696	0.167	0.985	0.141
	EMPHCN	0.092	0.845	0.261	0.986	0.178

**Table 4 biomolecules-12-01666-t004:** Top 10 related drugs for breast carcinoma predicted by the EMPHCN.

Disease	Rank	Candidate Drug	Evidence
Breast carcinoma	1	Topotecan	[44]
	2	Gemcitabine	[47]
	3	Carboplatin	[45]
	4	Bleomycin	[46]
	5	Cisplatin	[48]
	6	Hydroxyurea	[49]
	7	Methotrexate	[50]
	8	Melphalan	[51]
	9	Thiotepa	[52]
	10	Trabectedin	[53]

**Table 5 biomolecules-12-01666-t005:** Top 10 related drugs for Parkinson’s disease predicted by EMPHCN.

Disease	Rank	Candidate Drug	Evidence
Parkinson’s disease	1	Pergolide	[54]
	2	Metixene	[30]
	3	Orphenadrine	[56]
	4	Galantamine	[55]
	5	Donepezil	[57]
	6	Cabergoline	[58]
	7	Cyclobenzaprine	[59]
	8	Gabapentin	[60]
	9	Dantrolene	NA
	10	Mephentermine	NA

**Table 6 biomolecules-12-01666-t006:** Top five related diseases for Felodipine predicted by the EMPHCN.

Drug	Rank	Candidate Disease	Evidence
Felodipine	1	Angina pectoris	[61]
	2	Edema	[62]
	3	Cerebrovascular accident	[63]
	4	Diabetic nephropathy	NA
	5	Congestive heart failure	[64]

## Data Availability

https://github.com/hwh7301/EMPHCN (accessed on 1 January 2022).

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
