# Peer review of "Drug Repositioning Based on the Enhanced Message Passing and Hypergraph Convolutional Networks"

_biomolecules, 2022, doi:10.3390/biom12111666_

Round 1

Reviewer 1 Report

The authors presented a new method of drug repositioning based on the enhanced message passing and hyper graph convolutional networks. They have shown that their method performs better than other methods based on similar general principles such as: DRRS, SCMFDD, LAGCN, NIMCGCN, BNNR and DRHGCN.

My main criticism is that the present paper is very technical, compares the present approach to conceptually similar technically oriented  papers and doesn't give the reader a broader perspective.

It is not discussed if the present paper can use genomic, transcriptomic, and other omics and cell line data from an individual patient to prescribe him a specifically repurposed drug. Other drug repurposing methods that are based on the analysis of  pathways affected by affected by the genes whose expressions are increased or decreased, based on Connectivity Map (CMAP) are ignored.  

In my opinion the manuscript needs a major revision to address these problems.

Reviewer 2 Report

This is very well written in terms of spelling and grammar, although it should be noted that there are a number of mis-hyphenations that likely emerged from pasting text from one word processor to another.

Technically, the study is solid.  I should mention one semantic matter, though.  The abstract cites the use of 10-fold cross-validation as a measure for model robustness.  I would normally have considered this to be insufficient for the specific application, since the study relies on artificially 'standardized' data (e.g., as implemented by resources that aggregate information from multiple sources) which can suffer from indirect biases introduced by imputation or other data cleaning schemes.  Thus, I would normally have asked for a more stringent 5-fold cross-validation test, to ensure true predictivity... but that is really what is described in lines 440-451 (page 12).  So, I do *not* need to make that request, though I might suggest that the authors mention this fairly important validation check in their abstract and/or conclusions.

One other matter that puzzled me is that although the authors devote substantial space to describing the metrics and methods used for gauging similarities between drugs and between diseases, the authors do not describe any use of protein-protein similarities which, of course, are readily characterizable through sequence alignment, motif conservation, and so forth.  I suppose it would be possible to development meaningful drug repurposing models by using only established protein-protein and drug-protein data, but practicing pharma developers usually pay attention to simple protein homology assessments to try to perceive possibly problematic off-target effects arising from the designed drug also modulating other proteins that shared sequence or motif homology with the intended target.  Since one person's risk (off target consequences) is another person's reward (repositioning opportunities), I would have felt this this consideration might be at least worth mentioning for future augmentation.

Reviewer 3 Report

Current manuscript deals with a new drug repositioning method (EMPHCN), which involves the enhanced message passing and hypergraph convolutional networks (HGCN). Based on the importance of the topic the manuscript is ACCEPTABLE for the publication after the MAJOR REVISION of the following:

1.       Some important review articles need to be added (Nat Rev Drug Discov 18, 41–58 (2019) https://doi.org/10.1038/nrd.2018.168; J Cheminform 12, 46 (2020). https://doi.org/10.1186/s13321-020-00450-7).

2.       The authors provided a successful example of identifying drugs for breast carcinoma and Parkinson’s disease. And what about application of the herein reported method for some discontinued drugs, for, instance, recently discontinued by Astra Zeneca type 1 diabetes drug dapagliflozin, calcium channel blocking agent felodipine, etc.? Or Remdesivir, which was proposed for the treatment of COVID-19? Is it possible by using the herein reported method to predict the application of the above-mentioned drugs for the treatment of other conditions? Some examples would be helpful.

Round 2

Reviewer 1 Report

The revised version of the paper is significantly improved and can be published in Biomolecules.

Reviewer 3 Report

The authors have improved their manuscript according to my suggestions. I would recommend it for the publication in its current form.